# Physiology can predict animal activity, exploration, and dispersal

Nicholas C. Wu[1] & Frank Seebacher [1✉]

Physiology can underlie movement, including short-term activity, exploration of unfamiliar environments, and larger scale dispersal, and thereby influence species distributions in an environmentally sensitive manner. We conducted meta-analyses of the literature to establish, firstly, whether physiological traits underlie activity, exploration, and dispersal by individuals (88 studies), and secondly whether physiological characteristics differed between range core and edges of distributions (43 studies). We show that locomotor performance and metabolism influenced individual movement with varying levels of confidence. Range edges differed from cores in traits that may be associated with dispersal success, including metabolism, locomotor performance, corticosterone levels, and immunity, and differences increased with increasing time since separation. Physiological effects were particularly pronounced in birds and amphibians, but taxon-specific differences may reflect biased sampling in the literature, which also focussed primarily on North America, Europe, and Australia. Hence, physiology can influence movement, but undersampling and bias currently limits general conclusions.

[1] School of Life and Environmental Sciences, The University of Sydney, Sydney, NSW 2006, Australia. ✉email: frank.seebacher@sydney.edu.au

Movement is fundamental for animal ecology. Movement during foraging, for example, influences habitat use and interactions within local ecosystems[1]. At a larger spatial scale, dispersal drives colonisation of novel habitats and impacts biogeography[2]. Dispersal within meta-populations determines rates of gene flow between individual populations and influences genetic variation and adaptation to different environments[3]. For example, low rates of gene flow may decrease genetic diversity within single populations and limit the potential for selection. Increased dispersal could "rescue" isolated populations by increasing genetic diversity[4]. On the other hand, genetic variation between populations within a greater meta-population can increase resilience to environmental change of the meta-population as a whole via the portfolio effect[5]. Range shifts resulting from dispersal may also expose populations to more favourable environments and may thereby increase resilience to environmental change[6,7].

Movement at any scale, from activity within familiar environments to exploration of novel environments and dispersal, requires the motivation to move. Hence, movement comprises "initiation" consisting of the decision to move, "transience" denoting the actual movement, and "settlement" in a new location in the case of larger-scale movement such as dispersal[8]. Dispersal may be defined as displacement of animals from their origin that can have repercussions for gene flow[4].

Multiple environmental and biological factors such as resource availability, social interactions, and changes in habitat quality may contribute to the initiation of movement[9]. The motivation to move may be driven by inter- or intraspecific competition within the original distributional range, which excludes individuals from access to resources and stimulates the search for resources elsewhere[9]. Additionally, changing environmental conditions, such as changes in temperature, may cause unfavourable conditions in the original habitat and stimulate individuals to search for more favourable conditions. These dynamics may be influenced by environmental conditions experienced by previous generations and at early life history stages[10,11]. For example, mismatches between the parental environment and the actual environmental conditions experienced had negative influences on physiological performance and stimulated dispersal in guppies (Poecilia reticulata)[12]. Similarly, exposure of mothers to predators increased their daughters' tendency to disperse in the fish Gambusia affinis[13]. In the spider Cyrtophora citricola, the early natal environment influenced the dispersal behaviour of offspring[14]. Individuals vary in their tendency to initiate movement because each receives somewhat different information from the environment, and the speed and distance moved may depend on the physiological capacity of individuals[8,15]. Consequently, not all individuals in a population are likely to disperse[16].

The motivation to initiate movement may be driven by neuroendocrine processes that translate environmental stimuli to locomotor activity[17,18]. On the other hand, physiological characteristics can constrain movement after initiation. Energetics and aerobic energy (ATP) production are the most frequently recognised physiological constraints of movement[19–21], and metabolic rates may be linked to behavioural phenotypes that have a greater or lesser propensity to move[21–23]. Swimming, flight, and terrestrial movements such as running and walking rely on muscle-powered locomotion, and muscles require ATP for contraction and relaxation so that energetics could pose a strong constraint[24]. Additionally, calcium cycling and efficiencies in muscle power production[25–27] could influence muscle endurance and thereby dispersal[28]. Other physiological constraints include cardiovascular function. The capacity of the heart to pump sufficient blood to sustain exercise (cardiac scope) may be constrained by environmental conditions and thereby limit movement particularly under challenging conditions such as against high water flow[29].

Physiological characteristics typically vary between individuals within populations, and these differences may impact the tendency and extent of movement. For example, there was a three-fold difference in the metabolic cost of transport (i.e., the energy used to move a given mass for a given distance) among individual zebrafish, which influenced the distance individuals moved in an artificial stream[30]. Similarly, metabolic rates and locomotor performance can vary widely among individuals within species[31–33]. If movement relied on physiological capacities, it may be expected that the variation in physiological characteristics introduces differences in the tendency to move among individuals of the same populations. Such individual differences can have consequences for gene flow and genetic diversification within a species. For example, specific individual traits may be distributed unevenly among (meta)populations if these traits are associated with characteristic dispersal rates of individuals[34]. Hence, physiologically mediated differences in dispersal rates may also determine trait distributions among populations and possibly population success[34].

On the other hand, the movement itself may cause physiological differentiation within populations if it led to the separation between the expanding movement edge and the core of the distribution. As briefly discussed above, individuals with a greater tendency to move may have particular physiological characteristics, leading to core-edge differences. It is an interesting and as yet unresolved question whether any putative differences between individuals at the core and those at the edge are mediated genetically or epigenetically, or by a mixture of both[35]. In a species expanding into novel environments, conditions at the dispersal front may differ substantially from those of the core distribution. As a consequence, the phenotypes that are successful in environments at the range edges may be different to the most successful phenotype at the core of the distributions; these differences may arise because individuals with particular genetic make-ups or greater capacity for plasticity have greater fitness at the range edge[36,37]. Potential differentiation in physiological phenotypes between the core and edges may influence animal ecology through a variety of underlying traits, from disease resistance to social behaviour[38,39]. Physiology can also establish a causal link between changes in the environment and movement. For example, temperature and climate can have profound effects on physiology and movement patterns[40,41], and the sensitivity to these environmental changes may differ between individuals.

Our aim was to determine the current state of knowledge regarding the importance of physiological traits for animal movement, including activity within familiar environments, exploration of novel environments, and dispersal. We address two fundamental questions: (1) do physiological capacities of individuals promote or constrain movement ('Individual movement'), and (2) does dispersal drive physiological differentiation within populations ('Population range expansion'). We conducted two related meta-analyses of the published literature (total: 131 papers, 524 effect sizes, 95 species) to address these questions. Firstly, we analysed published experimental data that measured physiological traits in individuals and related these to movement by the same individual. Secondly, we analysed published studies that compared physiological traits between individuals at the distributional core to those at the range edge.

## Results

**Individual movement**. The final data set consisted of $k = 272$ effect sizes from 88 studies across 74 species (Supplementary Fig. S1). There was a geographical sampling bias, and most

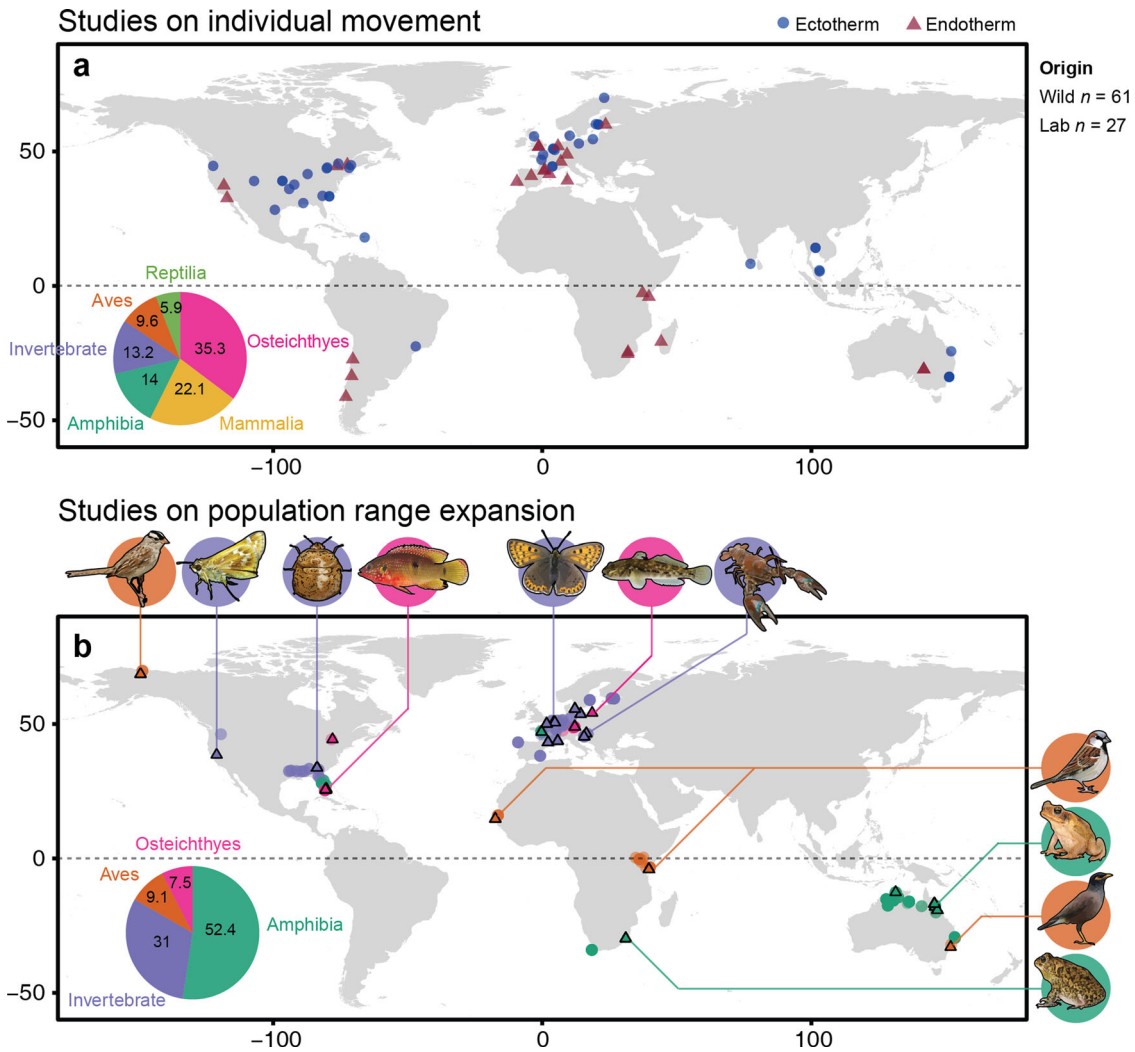

**Fig. 1 Geographical distribution where extracted studies were conducted. a** Locations of studies that investigated the relationship between physiology and dispersal propensity using wild animals, or where wild animals were collected for laboratory-based study. Data were coloured by thermal strategy (ectotherm as blue circle, and endotherm as red triangle). **b** Location of studies that compared physiology and locomotion between populations from the range core (triangles with black outline) and range edge (open circles). Data were coloured by taxonomic groups as shown in the inset pie chart (%). Example species listed from left to right, and top to bottom: *Zonotrichia leucophrys, Atalopedes campestris, Megacopta cribraria, Hemichromis letourneuxi, Lycaena tityrus, Neogobius melanostomus, Pacifastacus leniusculus, Passer domesticus, Rhinella marina, Acridotheres tristis* and *Sclerophrys gutturalis*. All artwork was produced by N. C. Wu, and animal images were based on photographs available under a Creative Commons licence (*Zonotrichia leucophyrus*, Wolfgang Wander, CC BY-SA 3.0; *Atalopedes campestris*, Charles T. Bryson, CC BY 3.0 us; *Megacopta cribraria*, Judy Gallagher, CC BY 2.0; *Hemichromis letourneuxi* Noel Burkhead, CC BY-SA 2.5; *Lycaena tityrus*, Robert Flogaus-Faust, CC BY 3.0; *Neogobius melanostomus*, Peter van der Sluijs, CC BY-SA 3.0; *Pacifastacus leniusculus*, Andreas Eichler and David Perez, CC BY-SA 4.0; *Passer domesticus*, Adamo, CC BY 2.0 de; *Rhinella marina*, Sam Fraser-Smith, CC BY 2.0; *Acridotheres tristis*, Sayan Dey, CC BY-SA 4.0; *Sclerophrys gutturalis*, Frank Teigler CC BY-NC 3.0).

studies on non-model species were conducted in Western Europe and North America (73.7%; Fig. 1a). Overall, physiology was positively associated with movement (grouping activity, exploration, and dispersal together) although there was some overlap of the 95% credible intervals with zero (Supplementary Table S3); note that credible intervals are the Bayesian equivalent of confidence intervals, and are interpreted in the same way. There was no strong effect of thermal strategy (endothermy or ectothermy), sex, age and sampling origin (Supplementary Table S3), and there was no evidence of sampling and publication bias (Supplementary Table S3). Heterogeneity in the data set was high ($I^2$ total = 97%; Supplementary Table S4) indicating that most variation remained unexplained.

Next, we analysed the effects of physiological traits on activity in familiar environments, exploration of novel environments, and dispersal separately. Only body condition, locomotor

performance, and metabolism had sufficient numbers of effect sizes for analysis. Of these, metabolism and condition did not have a pronounced effect (large overlap of credible intervals with zero), but there is a high level of confidence (marginal overlap of credible intervals with zero) that locomotor capacity had a positive effect on activity (Fig. 2; Supplementary Table S5).

We further subdivided the categories with the most effect sizes, metabolism and locomotor capacity (Fig. 3), to test their effect on movement (grouped activity, exploration and dispersal). There were positive effects of active metabolic rate, sprint speed and endurance, although the overlap with zero of the credible intervals reduce the confidence of the latter two results (Fig. 3; Supplementary Table S6).

Among different phylogenetic groups, the greatest positive effect of physiological traits was on activity in fish and invertebrates (Fig. 4; Supplementary Table S7). However, the

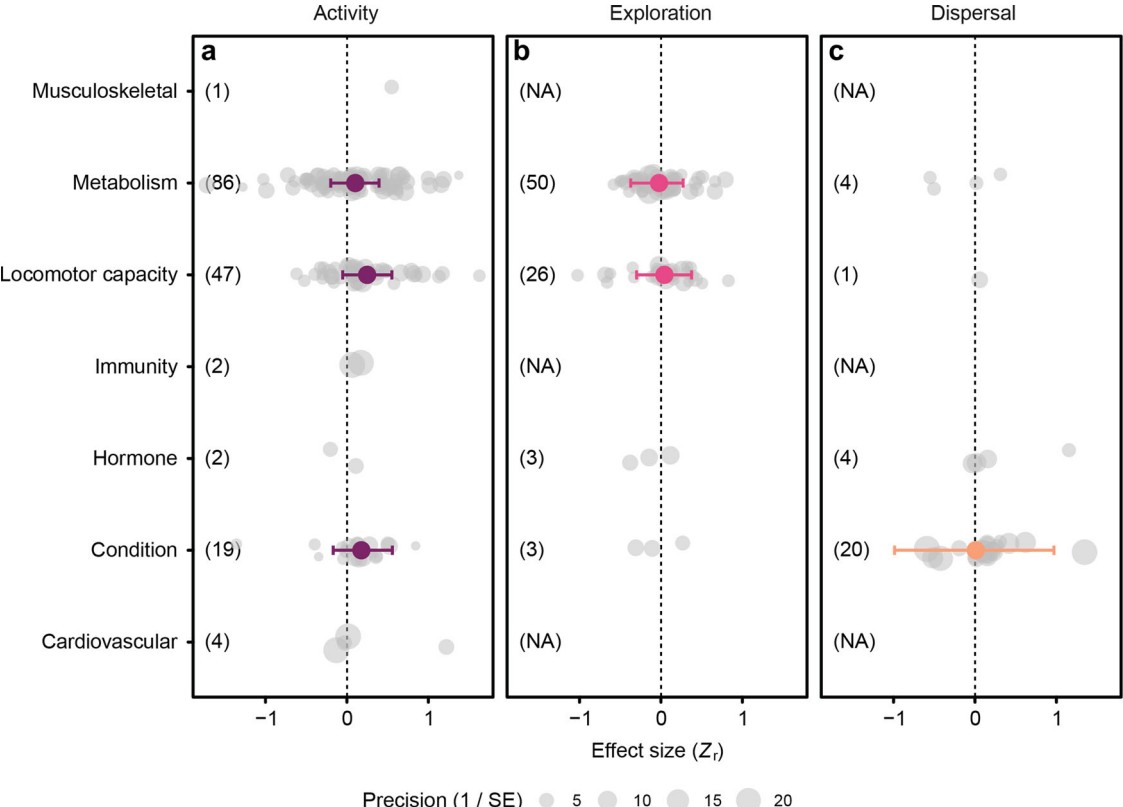

**Fig. 2 Effects of physiological traits on activity, exploration and dispersal. a** Activity in familiar environments. **b** Exploration of novel environments.
**c** Dispersal. The only effect with reasonable confidence was the effect of locomotor capacity on activity. Only traits with more than five effect sizes were
analysed. Numbers in brackets indicate a number of effect sizes (NA = not available). Grey points represent individual effect sizes, and the size of each
symbol indicates study precision (inverse of standard error). Effect sizes ($Z_r$) are estimates ±95% credible intervals.

overlap with zero of the credible intervals reduces the confidence
of these results, and there was considerable variation in the
number of effect sizes for each taxonomic group.

**Population range expansion**. The final data set comprised a total
of $k = 252$ from 43 studies across 16 species (Supplementary
Fig. S1), and 51.1% of studies extracted were conducted in
Western Europe and North America (Fig. 1b). Of those studies,
36 examined species expanding from their introduced range, and
seven with species expanding from their native range. The rate of
dispersal differed between dispersal modes, and animals that used
aerial locomotion had higher rates of dispersal than those using
terrestrial and aquatic locomotion (Fig. 5a; Supplementary
Table S8). The rate of dispersal also increased as the temperature
of the range edge increased relative to the range core (0.11 [95%
CI: 0.06–0.17]; Fig. 5b; Supplementary Table S8); in other words,
animals tended to disperse into warmer environments. Annual
rainfall did not predict the rate of dispersal (0.02 [95% CI: −0.01
to 0.005]; Supplementary Table S9). There was high confidence in
a positive relationship (95% CI: −0.02 to 0.14) between the time
since divergence and the magnitude in effect size, and the longer
the time since core-edge separation, the greater the effect size
(Fig. 5c; Supplementary Table S10).

Across all physiological traits, populations at the range edge
differed physiologically compared to the core range (0.25 [95%
CI: −0.02 to 0.52]; Supplementary Table S11). There was no
evidence of sampling and publication bias in our analysis
(Supplementary Table S11). Heterogeneity in the data set was
high ($I^2$ total = 99%; Supplementary Table S4). Among individual
physiological traits, the traits with the greatest confidence in
positive effect sizes (i.e., little or no overlap of credible intervals

with zero) were hormone (corticosterone) levels, immunity,
metabolism, and thermal tolerance, which were higher at the
range edge compared to the range core (Fig. 6a; Supplementary
Table S12). All taxa showed positive effect sizes, indicating
differences between core and edge, however fish and invertebrates
had a greater overlap of credible intervals with zero (Fig. 6b;
Supplementary Table S1).

**Discussion**
Physiology is undoubtedly important in enabling movement.
Muscles provide the power for movement, energy metabolism
supplies ATP, and the cardiovascular system delivers oxygen to
mitochondria. The importance of understanding the physiolo-
gical underpinnings of movement lies in assessing constraints
of movement. Physiological capacities can be limiting both
per se and as a result of environmental impacts[42,43]. Physio-
logical processes also interact and there may be bottlenecks
where limited capacity in one system constrains the function of
other systems and of the organisms as a whole[44]. Physiological
traits are inherently variable between individuals and species,
and if physiology predicted movement, physiological traits
could be used as indicators to predict the propensity for
movement and dispersal[19]. Such predictions would be valuable
to forecast responses to environmental change and to anticipate
invasiveness.

Our analysis of the current state of knowledge indicates that
physiological traits considered by existing studies have only
limited influences on movement by individuals. As expected,
active metabolic rate, sprint and endurance locomotor perfor-
mance were positively associated with movement. However, there
was quite a large variation in their effect sizes so the confidence in

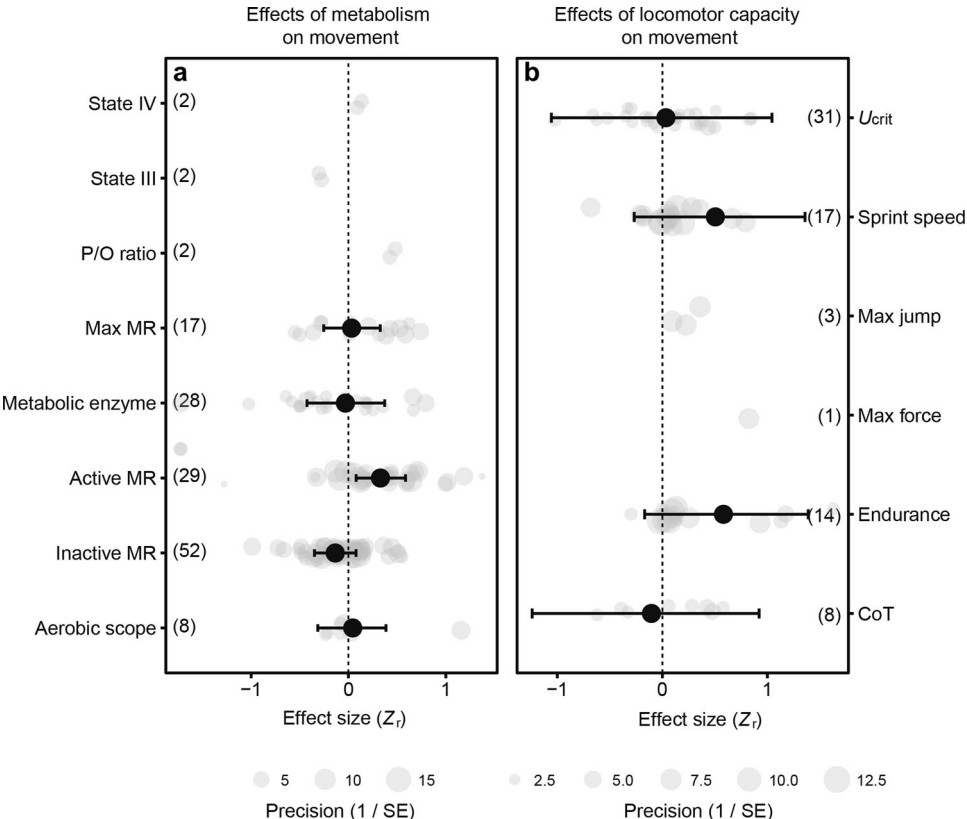

**Fig. 3 Effect of metabolic or locomotor traits on movement. a** Of the metabolic traits measured, active metabolic rate (MR) had a positive effect on movement. **b** Among locomotor traits, sprint and endurance had positive effects albeit with an overlap of 95% CI with zero. Activity, exploration, and dispersal were grouped together as 'movement', and only physiological traits with more than five effect sizes were analysed. Grey points represent individual effect sizes, and the size of each symbol indicates study precision (inverse of standard error). Effect sizes ($Z_r$) are estimates ±95% credible intervals.

their predictive power was limited, particularly for locomotor traits.

Interestingly, resting and maximal metabolic rates did not predict movement, which is contrary to the paradigm that high resting metabolic rates are associated with increased movement[45]. High resting metabolic rates indicate that individuals require relatively large amounts of food-derived energy to maintain cellular integrity and function. Consequently, it would be expected that individuals with relatively high resting metabolic rates also need to forage to a greater extent and therefore show a greater propensity to move[46,47]. However, the current literature indicates that this is not the case universally. The reason for this discrepancy may be that there is not necessarily a proportional relationship between resting metabolic rate and food intake. Instead, the quality and availability of resources and stored energy may decouple movement from resting metabolic rates[46,48]. High quality and at least temporary availability of abundant food would increase energy storage so that animals with relatively high resting metabolic rates would not need to forage consistently at a higher rate. Additionally, movement itself incurs costs and there are pronounced differences in the cost of transport—the energy used to move a unit of mass for a given distance—between individuals[30]. Individuals with a greater cost of transport may also move and forage less to reduce overall energy use[49].

It is possible that maximal metabolic rates constrain movement because lower maximal rates limit energy supply to muscles. For example, bank voles (*Myodes glareolus*) with greater maximal metabolic rates also moved greater distances, but basal metabolic rates were not correlated with movement[50]. It may be expected that maximal metabolic rates constrain movement if the required

movement occurs at near maximal capacities, such as a high speed or under difficult environmental conditions[51]. Note, however, that oxygen consumption is not necessarily correlated with ATP production, because mitochondrial ATP production efficiency can vary between contexts and individuals[48,52]. Hence, mitochondrial function could decouple maximum metabolic (oxygen consumption) rates from movement patterns. Other physiological traits, such as cardiovascular capacities, may also constrain movement in high-demand environments. For example, movement by salmon against river currents requires maximal locomotor performance which is constrained by cardiovascular capacities, particularly under warm conditions[29]. Our analysis showed, however, that there are too little data available in the literature to draw general conclusions about constraints of movement by the cardiovascular system.

Under most circumstances, animals do not move at maximal capacity but at a slower speed and in more benign conditions. Lower than maximal speeds can be advantageous because the dexterity of animals decreases as speed increases so that animals are more vulnerable to accidents[53,54]. Additionally, processing signals from the environment such as availability of resources, topographical information, or potential dangers decreases as speed increases[55]. Not surprisingly, therefore, animals rarely move at maximal speed even when escaping predators[56]. Hence, it would be expected that maximal locomotor capacities are not necessarily strong predictors of movement patterns. Maximal capacities are useful measures to determine what animals are capable of achieving when challenged to a high degree or to identify how physiological systems are affected by external influences such as changes in temperature. Most of the data on

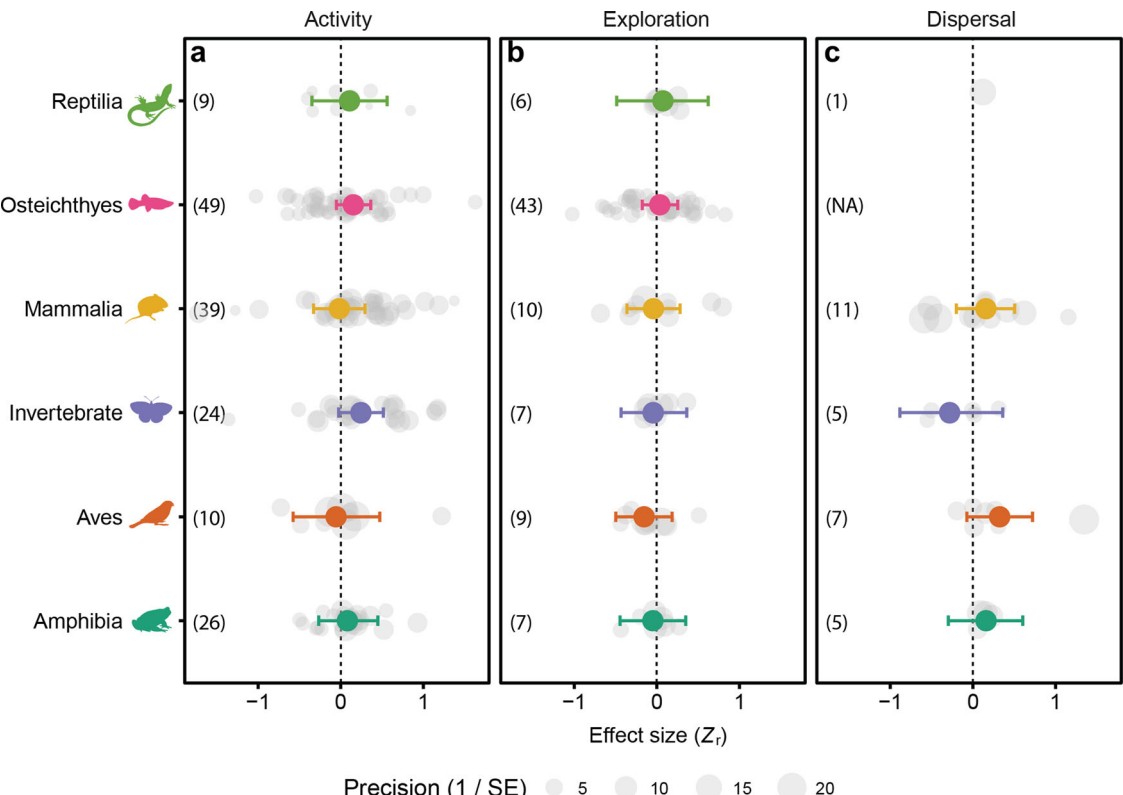

**Fig. 4 Effects of physiological traits on the movement of different taxonomic groups.** In this analysis, physiological traits were lumped together to determine their overall effect on **a** activity, **b** exploration and **c** dispersal of different taxa. Only taxa with more than five effect sizes were analysed. Effects of physiological traits were generally weak (large overlap of credible intervals with zero), and the strongest effects were on the activity of fish (Osteichthyes) and invertebrates. Numbers in brackets indicate the number of effect sizes for each taxonomic group (NA = not available). Grey points represent individual effect size and the size of each symbol corresponds to study precision (inverse of standard error). Effect sizes ($Z_r$) are estimates ±95% credible intervals.

activity and exploration in our analysis were derived from laboratory studies. The motivation for movements such as exploration of unfamiliar environments may differ between laboratory settings and the field[57]. Movement speed and underpinning physiological (e.g., metabolic) processes may thereby also differ between field and laboratory settings. There are currently not sufficient field studies on activity and exploration for formal comparisons, and this would be an interesting avenue for future research.

"Movement" is of course not a single entity or single trait but encompasses a range of purposes, distances, and durations. The physiological dimension of larger-scale movement such as dispersal where animals are displaced more or less permanently from their origin is relatively poorly documented. For obvious logistic reasons, more studies focussed on relatively short movements in experimental arenas. These studies are valuable because they can reveal physiological dimensions of movement (e.g., food supply and metabolic rates discussed above), but it is likely that longer dispersal would require different physiological inputs than relatively short-term movement in an arena. Our analysis comparing physiological characteristics of individuals within the core of distribution to those at the range edge indicates that dispersing individuals can have very different physiological make-ups, which are not seen in smaller-scale movements.

Individuals at the range edge showed increases in metabolism, immunity, and hormone levels compared to those at the core of the distribution. There was also an increase in thermal tolerance and locomotor capacity at the range edge although credible intervals indicated lower confidence in these effects. Increases in

energy metabolism and locomotor performance are likely to be more important in larger-scale movement as discussed above. Increased immunity would reduce morbidity and increase survival in novel environments that may harbour novel pathogens or a novel combination of pathogens[58]. Glucocorticoids (corticosterone) were the only hormones measured in core-range comparisons. Glucocorticoids have multiple functions that pertain to movement, including regulating energy metabolism[59], diel rhythmicity of locomotor activity[60], behaviour[61] and responses to environmental signals[62]. Interestingly, the overall trend in the direction of movement of animals in our analysis was from cooler towards warmer environments. The increased thermal tolerance of individuals at the range edge may be associated with that trend. It is not clear whether there is a cause-and-effect relationship that can explain this trend, but if it were robust—and considering the geographical bias in the data (see below) this is not a given—climate change could disrupt dispersal patterns if warmer environments exceeded heat tolerances.

Differences in physiological characteristics between core and edge may have implications for gene flow and adaptation. If physiological traits were determined genetically, the increased likelihood of individuals with particular physiological characteristics to disperse would lead to genetic divergence between core and edge[63]. Physiological differences would promote genetic differences because particular physiological phenotypes, and hence genotypes, would segregate between core and edge. Alternatively, differences in physiology between core and edge could reflect plasticity. Either all individuals within the general population have the same capacity for plasticity and physiological

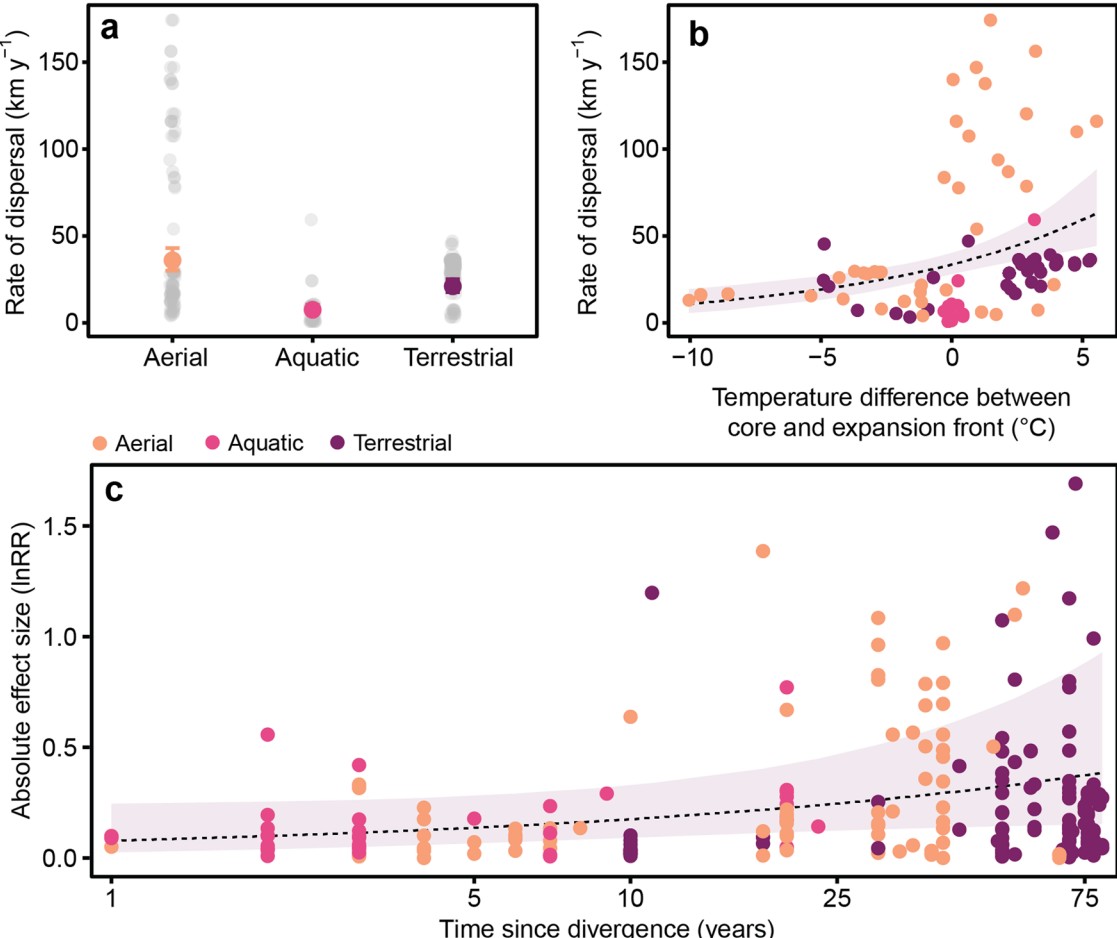

**Fig. 5 Relationship of temperature on dispersal rate, and the time since populations diverged on the magnitude of change in physiological traits. a** Differences in the rate of dispersal (km y$^{-1}$) between dispersal modes, and **b** the relationship between the annual mean air temperature difference between the range core and dispersal front (°C) and the rate of dispersal. **c** Absolute effect size (lnRR) showing the influence of time since populations diverged on the magnitude of change in physiological traits. Data in the plot are presented as average model estimate ±95% CI, and grey points represent raw data. Data in plots **b** and **c** are represented as the regression estimate ±95% CI, and individual data points are coloured by dispersal mode.

differences simply reflect different environmental inputs at core and edge. Or individuals with greater capacity for plasticity are more successful at dispersing so that there is a gradient in capacity for plasticity between core and edge. In the latter case, there may also be differences in the resilience to environmental change, and individuals at the core may be more likely to be negatively affected and hence more vulnerable to extinction, compared to individuals at the edge[64]. Such a (hypothetical) gradient in plasticity could have pronounced influences on the dynamics and genetic structure of populations. We found a tendency that physiological differences increased with time since the separation between core and edge, which would indicate that phenotypic changes are mediated by processes that are slower acting than reversible acclimation, such as transgenerational epigenetic effects or adaptation if selection played a role.

Similar to findings from other meta-analyses, our analysis of the literature unveiled a strong geographical bias[65]. By far the most studies were conducted on organisms from North America, Europe and Australia for core-edge comparisons. Additionally, the coverage of different physiological traits and taxonomic groups is sparse, and investigations of the physiological basis of movement by individuals is almost entirely focused on locomotor capacity and metabolism. Hence, the discussion above may summarise the current state of knowledge, but it is unlikely to represent true biological generalities. Clearly, future work needs

to focus on broadening the scientific base, particularly because the geographical areas worst affected by climate change are also the least sampled[65]. Although all major groups of vertebrates are represented in the literature, their coverage is sparse and the species studied do not represent vertebrate classes as a whole. Invertebrates are even more poorly represented, particularly considering their much greater diversity. Other outstanding questions include whether physiological differences between the core and edge of distributions are the cause or effect of dispersal, whether physiological differences are mediated genetically or reflect plasticity, and how these dynamics will affect population structures.

## Methods

**Literature search and effect size calculation**. We followed PRISMA guidelines[66] in the design and analysis of this study (Supplementary Fig. S1). We searched the primary literature in the Web of Science (subject area: Ecology, Zoology, Marine Freshwater Biology, Evolutionary Biology), ScienceDirect (subject area: Agricultural and Biological Sciences) and Scopus (subject area: Agricultural and Biological Sciences) on the 10 March 2021. The search terms are given in Supplementary Fig. S1. Title and abstract screening were conducted in Rayyan[67].

*Individual movement*. The aim of this analysis was to determine whether physiological traits are related to movement by individuals, and we analysed studies that report a measure of movement and physiological trait(s) in the same individual of any species that uses muscle-powered locomotion for activity, exploration, or

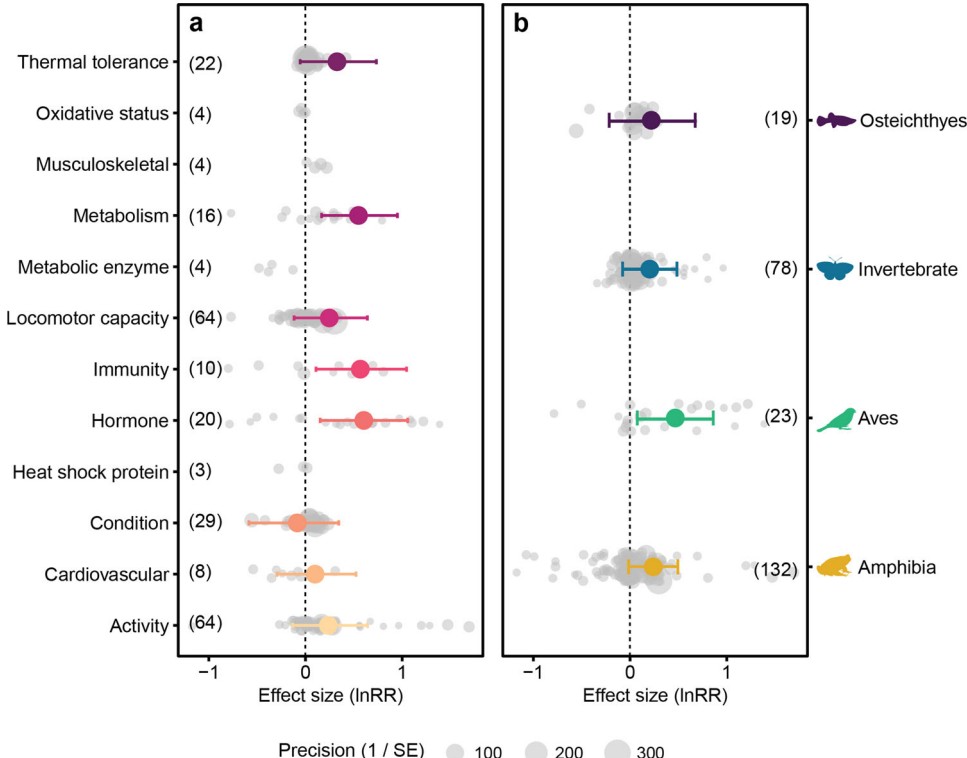

**Fig. 6 Differences in physiological traits between population core range and dispersal front. a** Effect sizes showing differences between the range core and edge in physiological traits across all taxa, and **b** overall effect of all (lumped) physiological traits on different taxonomic groups. Only traits or taxa with more than five effect sizes were analysed. Positive direction indicates populations at the range edge have higher physiological capacity. Numbers in brackets indicate the effect size for each categorised trait or taxon. Grey points represent individual effect sizes, and the symbol size of each individual effect size corresponds to study precision (inverse of standard error). Effect sizes (lnRR) are estimates ±95% credible intervals.

dispersal (see also Supplementary methods, data exclusion criteria). Studies were included if all of the following criteria were met:

(1) The study reports measurements of movement by individual animals such as 'activity' and 'exploration'. Most experimental studies investigated dispersal 'potential' or 'tendency' as activity in a familiar environment or exploration of a new environment, and therefore we included these responses as separate categorical groups[68]. We included both laboratory and field studies; in the event, most studies measuring 'activity' and 'exploration' were laboratory-based (85%), while dispersal was measured primarily in the field (90%). We defined 'activity' as a movement in a familiar environment measured as distance moved or the number of movements per unit of time. 'Exploration' was defined as movement in an unfamiliar environment reported as distance moved per unit time, latency to enter an unfamiliar environment, a number of times re-entering an arena, a number of unique zones visited, or principal components related to these measured movement parameters. 'Dispersal' was defined as a muscle-powered, non-returning movement away from breeding home ranges that are completed within one generation[15]. We, therefore, excluded migration, which is a returning movement between feeding and breeding home ranges, and passive dispersal through dispersal vectors such as water- and wind-currents, or transport by other organisms.

(2) The study measured physiological responses associated with activity, exploration or dispersal. Most effect sizes stem from measures of metabolism ($k = 106$), locomotor performance ($k = 66$), and body condition ($k = 40$), and a full list of physiological measures is given in Supplementary Table S1.

(3) The study presented correlations between movement (criterion 1) and a physiological trait (criterion 2) at the individual level or presented inferential statistics ($t$, $F$, $\chi^2$) that allowed estimation of the correlation indirectly (Supplementary methods). Studies that only presented categorical groups of dispersal phenotypes (i.e., residents or philopatric vs. dispersers) were not included unless raw data of physiology and movement were provided.

A detailed description of the data collected are given in Supplementary Table S1, and exclusion criteria are provided in Supplementary Methods.

Forward (papers that cite the original study) and backward (previous papers that the original study cited) searches were obtained from Google Scholar until 10th May 2021. We also searched for relevant articles from review papers on physiology and movement[8,21,69–71]. For all studies that met our criteria, we

extracted the correlation coefficient ($r$) and sample size ($n$) directly from the study or from figures via the *metaDigitise* package in $R$[72], and calculated the Fisher $z$-transformed effect size (Eq. 1) and sampling variance (Eq. 2) as follows[73]:

$$Z_r = \frac{1}{2}\ln\left(\frac{1+r}{1-r}\right) \qquad (1)$$

$$v(Z_r) = \frac{1}{n-3} \qquad (2)$$

For studies that did not present $r$, or showed appropriate figures to extract $r$, we obtained and transformed inferential statistics ($t$, $F$, $\chi^2$) to $r$ using standard conversions (Supplementary methods).

*Population range expansion.* In this analysis, we compared differences in the physiological characteristics between individuals from an established range core to their expanding range edge. We included studies that fulfilled all of the following criteria:

(1) It reported means, sample sizes ($n$), and variation as either standard deviation (SD), standard error (SE) or confidence intervals (95% CI) for both the range core group (central/core range, site of introduction) and range edge group (dispersal/invasion front);

(2) Reported measurement of at least one physiological response (Supplementary Table S2).

We also extracted information about the year when the populations at the range core and range edge were established, and the distance between the core and edge. Four studies did not report the time of the establishment of the core range, which we did not include in this analysis. The rate of dispersal (km y$^{-1}$) was calculated from the distance between the core and the edge, and the time since divergence. Differences in physiology between the population core and dispersal front were calculated as the natural log-transformed response ratio (lnRR; Eq. 3) and the sampling variance (Eq. 4) was calculated as[74]:

$$\ln RR = \ln\left(\frac{\bar{x}_F}{\bar{x}_C}\right) \qquad (3)$$

$$v(RR) = \frac{(\mathrm{SD}_F)^2}{N_F \bar{x}_F^2} + \frac{(\mathrm{SD}_C)^2}{N_C \bar{x}_C^2} \qquad (4)$$

where $\bar{x}_F$, $SD_F$ and $N_F$ represent mean responses, SD, and sample size for the range edge, respectively, while $\bar{x}_C$, $SD_C$ and $N_C$, represent the same parameters for the range core.

For the population range expansion dataset, we also extracted mean air temperature (°C) and precipitation (mm yr$^{-1}$) differences between the core and range edge directly from the study and categorised the medium of dispersal (aerial, aquatic and terrestrial). Note that all studies that reported temperature data measured these at the time of conducting fieldwork. For studies that did not include air temperature and rainfall, we extracted yearly mean air temperature and precipitation from the Global Climate Extractor in NicheMapper (http://niche-mapper.com/apps/climate/index.html) using the longitudinal and latitudinal coordinates from each study site. The earliest study in our data set was published in 2002, and climate change may have caused some divergence in temperatures between then and now.

For both 'individual movement' and 'population range expansion' datasets, we also included the following moderators: taxonomic information, thermoregulation strategy (ectotherm or endotherm), sex (male, female, mixed), age (juvenile, adult) and origin (wild-caught, captive-raised; the "origin" moderator is relevant for the 'individual movement' analysis only, and refers to common laboratory animals such as zebrafish). Where possible, we noted the geographical origin of study animals (except when these were model species distributed throughout the world, such as zebrafish) to describe potential sampling bias. In both analyses, effect sizes were coded so that positive values indicate an increase in both dispersal and physiological performance. For example, a decrease in critical thermal minima (CT$_{min}$) indicates an increase in tolerance to colder temperatures. Therefore, a decrease in CT$_{min}$ was coded to positive values. Latency to enter a novel area, and cost of transport were also reversed as a shorter latency indicates faster time to enter the novel environment, and lower cost of transport indicates more efficient energy use for movement.

To correct for phylogenetic nonindependence, we built phylogenies using the Open Tree Taxonomy nomenclature format and retrieved the phylogenetic relationships from the Open Tree of Life[75] using the *rotl* package in R[76]. Polytomy was accounted for via randomisation using the function 'multi2di', and branch lengths were estimated using the function 'compute.brlen' from the *ape* package[77]. The generated trees (Supplementary Figs. S2 and S3) were converted to a phylogenetic relatedness correlation matrix for subsequent analysis.

**Statistics and reproducibility**. All data were analysed in a Bayesian framework using the *brms* package[78]. All models were assigned default or weakly informed priors[79]. As variation in responses to the predictors (σ) can only be positive, we used a half-Cauchy prior with a location of zero and a scale of 1. For each model, we constructed four chains with 10,000 steps per chain, including 5,000-step warm-up periods, so a total of 20,000 steps were retained to estimate posterior distributions (i.e. $(10,000 - 5000) \times 4 = 20,000$). Adapt delta was set at 0.999 to decrease the number of divergent transitions and the maximum tree depth was set to 20 when the depth of tree evaluated in each iteration was exceeded. The degree of convergence was deemed as achieved when the Gelman–Rubin statistics, $\hat{R}$[80] was 1. Data were presented as mean posterior estimates ±95% credible intervals (95% CI); note that credible intervals are the Bayesian equivalent of confidence intervals, and are interpreted in the same way.

*Individual movement*. We first constructed a phylogenetically corrected meta-analytic model to examine the overall effect size of pooled data for activity, exploration, and dispersal. The model included four random factors to assess non-independence in the data[81]: (a) species identity, to account for similarities of effect sizes within the same species; (b) phylogeny, to account for similarity due to common ancestors; (c) study ID, to account for multiple effects per study; and (d) observation (effect size) level random effect, which is equivalent to the residual term in a normal linear model. We included publication year and effect size standard error [$\sqrt{\nu(z_r)}$] as covariates to account for time-lag and publication bias[82]. We calculated an extended heterogeneity ($I^2$) statistic to partition total heterogeneity ($I^2$ total) into within-species variance ($I^2$ species), phylogenetic variance ($I^2$ phylogeny), study ID variance ($I^2$ study) and residual variance ($I^2$ effect size)[83]. We visualised publication bias via funnel plot in Supplementary Fig. S4.

To determine if there were differences between physiological traits, or between different taxonomic groups, we implemented a model with physiological traits or with taxonomic groups as the main predictors, respectively. We analysed differences between physiological traits (trait model) and taxonomic groups (taxon model) for activity, exploration, and dispersal separately. We excluded the intercepts from the models to allow the estimation of coefficients for every factor level. We only analysed traits and taxa with five or more effect sizes[81]. For the taxon model, we ran a non-phylogenetic, multi-level model (excluding phylogeny as a random effect) as related species are grouped categorically for each assigned taxon group.

We examine individual traits in the metabolism and locomotor categories because there were sufficient numbers of effect sizes to allow more detailed analyses, which was not the case for the other categories (Supplementary Table S1). Within the metabolism category, we grouped metabolic rate into three groups: (1) non-active metabolic rates, grouping basal, resting, and standard metabolic rates; (2) active metabolic rate, where the metabolic rate was recorded in animals moving voluntarily and averaged over a day of recording such as routine and field

metabolic rate; (3) maximum metabolic rate, where the highest metabolic rate was measured either during forced exercise within a fixed time interval or immediately post-exhaustion.

*Population-level range expansion*. We constructed a phylogenetic, multi-level, meta-analytic model. As above, we tested for effects of physiological traits or taxonomic groups as main predictors, using these categories as main predictors. All models included four random factors as above (except the taxon model), with publication year and the square root of the inverse of effective sample size ($\sqrt{1/\tilde{n}_i}$) as covariate terms, which account for time-lag and publication bias. The effective sample size was preferable here because it accounts for unbalanced sampling between groups[82]. The inverse of effective sample size ($1/\tilde{n}_i$; Eq. 5) was calculated as

$$\frac{1}{\tilde{n}_i} = \frac{N_C + N_F}{N_C N_F} \qquad (5)$$

We also calculated the model's heterogeneity and visualised publication bias as above (Supplementary Fig. S4).

We tested if the rate of dispersal was influenced by environmental temperature or precipitation by constructing a regression model with the temperature or precipitation difference between the range core and range edge as the main predictors, and with dispersal mode as an interactive term. Lastly, we examined the effect of the time since the range edge diverged from the range core on absolute effect sizes. We constructed a regression model with time diverged (years) as the main predictor, with dispersal mode as an interactive term.

**Reporting summary**. Further information on research design is available in the Nature Research Reporting Summary linked to this article.

## Data availability
All datasets generated and analysed during the study are available on the GitHub repository: https://github.com/nicholaswunz/dispersal-meta-analysis[84]

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

## Acknowledgements

This research was funded by the Australian Research Council Discovery Grant (DP190101168) to F.S.

## Author contributions

F.S. conceived the ideas and wrote the paper, N.C.W. extracted and analysed data and prepared Figures, F.S. and N.C.W. edited and approved the paper.

## Competing interests

The authors declare no competing interests.
