## [Transparent Peer Review File · Communications Biology]

Reviewers' comments:

Reviewer #1 (Remarks to the Author):

The authors should be commended on a robust study, with excellent result presentation and a well written document.

the first analysis to assess the link between physiology and movement, is well established, but I presume was used to demonstrate suitability of the methods used.

The second analysis assessed if physiology differed between animals within the core or edge of the species their geographical ranges.

This is perhaps the more interesting analysis but after reading the paper multiple times I could not work out how you were able to compare the core and range effects upon physiology and location from the same species across multiple studies, which I presume used different sampling methods. Where these findings comparable?

This concept of differing physiology around core and edge effects has been demonstrated a few times now in single species, where the sampling methods were comparable. for example cane toads in Australia.

Reviewer #2 (Remarks to the Author):

BRIEF SUMMARY OF THE MANUSCRIPT

The authors conducted two separate meta-analyses to link physiology with ecology via animal movement, broadly defined.

The first meta-analysis focused on the link between physiological traits (measures of metabolism, locomotor performance, immunity, etc.) with animal movement (activity in a familiar environment, exploration in a novel environment, and dispersal). It resulted in limited support for effects of some physiological traits on movement.

The second meta-analysis examined whether individuals from the core of their range vs. the edge of their range differ in physiology. It provided evidence for divergence of numerous physiological traits, including some that would be expected to have ecological effects. Additionally, the degree of divergence is positively related to the time since divergence. It is not clear the extent to which this divergence might be the result of genetic differentiation vs. phenotypic plasticity. This second meta-analysis also explored whether temperature or precipitation might differ systematically between range core and edge, finding evidence for warmer temperatures at range edges (with some caveats/limitations).

OVERALL IMPRESSION

The authors have chosen an interesting and important topic with implications for a variety of fields. I'm always excited to see connections between individual-level traits and population-level phenomena!

As with many interesting and important topics, this one presents some inherent challenges (not insurmountable).

First, this work will likely interest readers in disparate fields, including physiology, population genetics, behavioral ecology, and community ecology. Researchers from these fields have very different knowledge bases, and they don't necessarily share a vocabulary. They likely have different ideas of what one usually means by "movement"! Thus, the authors must work hard to make sure they communicate clearly with all parts of the potential audience, which involves clearly defining/explaining terms and ideas that might not need defined/explained in a paper targeting a narrower audience. In my specific comments, I aimed to point out places where the authors could expand or improve their explanations to reach their broad audience.

Second, many researchers who would be interested in these results are not intimately familiar with meta-analyses or Bayesian statistics. As the manuscript is currently written, such researchers will likely have trouble understanding how to interpret some results, and could use more guidance from the authors. In my specific comments, I have indicated some places where the authors could provide such guidance.

The authors can certainly rise to these challenges with some revisions.

Finally, please note that I have never conducted a meta-analysis, nor am I intimately familiar with the mechanics of meta-analyses. I also have only a passing familiarity with Bayesian statistics. Therefore, I cannot evaluate the statistical analyses or the validity of claims based on those analyses. Therefore, the editor should ensure that another reviewer who does have experience with meta-analyses and Bayesian statistics has critically reviewed the methods and results sections.

SPECIFIC COMMENTS

To aid authors in preparing their response, specific comments are numbered consecutively (SC 1, SC 2, etc.), and most refer to line numbers.

TITLE

SC 1: The title given in the Supplementary Information document (Does physiology predict animal movement: exploration, dispersal, and invasiveness?) differs slightly from the title given in the main document (Does physiology predict animal movement: activity, exploration, and dispersal?) Either way, the punctuation makes it a bit odd to read, in my opinion. If other reviewers have the same impression, you may consider a changes (e.g., "Does physiology predict animal movement? A meta-analysis of physiological effects on X, Y, and Z" or something along those lines). It also seems to me like the title alludes to the first meta-analysis but not the second; authors can decide if that matters to them.

ABSTRACT AND INTRODUCTION

SC 2; Lines 34-46: Although this paragraph isn't labelled "Abstract," it reads as one and I'm going to interpret it as such. It sounds like your meta-analyses produced interesting findings, but this paragraph currently doesn't convey enough information for the reader to really "get" the context or importance of your findings – some things don't make sense at their current level of detail (I'll give specific examples below). Remember, the abstract is the only part of the paper that many people will read, so you want to make sure it conveys what's interesting and/or important about your findings (including why they are interesting and/or important). It would help if you drew more connections among ideas.

SC 3; Line 34 (and throughout): The word "movement" can convey many different ideas, and it's not

immediately clear to readers which one(s) you are focused on (moreover, readers from different fields will have different initial thoughts on what you might mean). Locomotion? Dispersal? Home range size? Daily movement distance? It would help to be more precise in your language. You don't need to erase all use of the word "movement" (for example, in lines 58-59 it seems appropriate and well-explained), but every time it appears, consider whether replacing it with a more precise word or phrase would improve communication. In the abstract, in particular, I think some replacements would help. The reader has to get pretty far into the paper before getting an explanation of what exactly "movement" means in the context of your analysis, and even then, it's not as clear-cut as I think it could be (additional specific comments provided below).

SC 4; Line 39: Similar to the previous comment, when you say "individual movement" do you mean "daily movement distance," or something else?

SC 5; Line 39: I notice a marked difference in the firmness/confidence of your conclusion in the abstract ("We show that locomotor performance and metabolism affected individual movement") vs. in the results ("There were positive effects of active metabolic rate, sprint speed, and endurance, although the overlap with zero of the credible intervals reduce the confidence of the latter two results"; lines 139-142) and discussion ("there was quite large variation in their effect sizes so that the confidence in their predictive power was limited"; lines 183-184). You should be careful to avoid overstating your results in the abstract, especially since that is the only part some people will read.

SC 6; Lines 38-41: You list several traits (locomotor performance, metabolism, individual movement, corticosterone, immunity) that differ between the range core and edge. How do you think these traits are connected, if at all? Do you (or others) hypothesize causal relationships? Correlations due to some underlying factor? Coincidence? If you can include a sentence or two to expand on these ideas, the abstract would become more compelling.

SC 7; Lines 41-42: Any idea why physiological differences would be more pronounced in birds and amphibians? Again, the abstract will be more compelling if you can go beyond a raw presentation of your findings.

SC 8; Lines 43-44: The trend for dispersal to occur in the direction of warmer environments has some major caveats/limitations. You point out some in the discussion, and I have some additional thoughts, detailed later in this review. Given these caveats/limitations, I think you should be much more careful about how you present this result in the abstract (and maybe even whether you should present it, especially given that it seems slightly tangential to the core focus of your paper).

SC 9; Lines 45-46: Can you be more specific? What traits (or types of traits) and geographic regions are currently underrepresented in the literature?

SC 10; Line 48: Careful with the word "defines." Although foraging is important for habitat use and interactions within ecosystems, other things matter, too (for example, finding mates or avoiding predators).

SC 11; Lines 63-64: "Information they receive from their environment" is just one of many reasons individuals may vary in their tendency to initiate movement. You get more into the initiation of movement in the next paragraph (lines 67-71); some minor re-organizing would help here. Maybe remove this content from the current paragraph and work it into the next?

SC 12; Line 64: What exactly do you mean by "movement per se"? Movement, broadly defined? The "per se" makes me wonder if you have something special in mind that I may or may not be getting.

SC 13; Lines 63-66: "Individuals vary in their tendency to initiate movement" ... "Consequently, there is substantial variation between individuals in their tendency to move" – this is a circular. I

recommend re-working these sentences (some re-working would probably happen anyway, if you take my suggestion of moving this idea into the next paragraph).

SC 14; Lines 74-76: Can you be more specific, maybe by providing an example? This sounds interesting, but not everyone in your potential audience will automatically know what you are talking about. Also, since it is a general statement, you might want to cite one or two more studies beyond just the one on guppies.

SC 15; Line 82: "Movement relies on muscle-powered locomotion" – many plant seeds disperse despite their lack of muscles, and planktonic organisms also move without muscle-powered locomotion. Perhaps specify which taxa/circumstances you are talking about when making such broad statements?

SC 16; Lines 85-86: Can you provide some explanation of what you mean by "cardiac scope" for readers who are not physiologists?

SC 17; Lines 88-90: "For example..." I can't put my finger on why, but this statement feels incomplete, as if you stopped partway through a thought (even though it is a grammatically complete sentence). Maybe read it out loud to yourself to see if you get the same feeling (and revise, if you do)?

SC 18; Lines 91-93: Given that "tendency to move" (in any sense of the word "move") relies on behavior as well as physiology, and behavior is more labile than physiology, then we might expect even more intraspecific variation in "tendency to move" than we see for the physiological traits discussed earlier in the paragraph. (You may or may not want to address this point.)

SC 19; Lines 96-98: "Dispersal success" seems like a subtly different concept from "tendency to move" or "dispersal rates" mentioned earlier in the paragraph (an animal might have a high "tendency to move," but it may not succeed in dispersing, or vice versa, for a variety of reasons that may or may not be related). Re-read to see if you are communicating what you mean to communicate, or if you need to edit or expand on any of these ideas.

SC 20; Lines 106-108: Perhaps streamline this sentence by cutting a clause: "Potential differentiation in physiological phenotypes between the core and edges may influence community ecology through a variety of underlying traits, from disease resistance to social behaviour."

SC 21; Line 113: Here, it would be especially helpful to precisely communicate what you mean by "movement," since this is where you define the scope of your study.

MATERIALS AND METHODS

SC 22; Lines 289-292: Individuals of what taxa? All animals? All vertebrates? A reader can currently glean the answer from figures and supplemental materials, but it should also be mentioned in the main text. This would be a good place to refer the reader to Figure 1.

SC 23; Lines 293-304: Is criterion (1) your definition for "individual movement"/"movement by individuals"? If so, can you make that explicit? If not, can you change something about this paragraph or the preceding one to clarify? (The answer to this question was clearer the second time I read the manuscript, but it needs to be crystal clear on the first read.)

SC 24; Lines 293-304: You make no mention of how many studies were lab-based vs. field-based. I suspect many (most?) experiments were lab-based, which could have implications for interpretation. Behaviors displayed in lab assays may or may not measure what we think they measure, and they

may or may not correlate with behavior in the field. At a minimum, it would be nice for the methods to indicate whether both categories of study were included, and for the discussion to touch on some of the implications for our interpretation of the results, assuming lab-based studies were included. Here's one recent paper on the topic:

Mouchet A. and N.J. Dingemanse. 2021. A quantitative genetics approach to validate lab- versus field-based behavior in novel environments. *Behavioral Ecology* 32:903–911.

SC 25; Lines 305-306: Even if you don't have space to list all measures of physiology in the main text, readers should get some idea of traits considered without having to consult the supplementary material. One possibility would be to say something like "measures of physiology included are given in Supplementary material Table S1; they include traits related to metabolism (15 total), locomotor capacity (6 total)..."

SC 26; Line 332: What do you mean by "recent arrival"? Is this somehow different from a dispersal/invasion front?

SC 27; Lines 335-336: How often do researchers have information on the year when the populations at the range core were established? I would expect this information to be readily available only for species that have been introduced to new locations and then expanded their introduced range. I would be surprised to see this information for a population in the core of a species' native range.

SC 28; Lines 335-336: It is not clear whether all the studies included in this second meta-analysis were of organisms expanding the size of their introduced range (such as cane toads in Australia), or whether some studies may have been of organisms that are expanding out from their native range (such as cowbirds in North America). It would be nice to know the proportion of each type of study included in the analysis. Since *Communications Biology* relegates Materials and Methods to tiny text at the end of the paper, you may consider clarifying this point elsewhere, as well.

SC 29; Lines 354-355: I'm concerned about the inclusion of data from captive-raised individuals, given phenotypic plasticity of many traits. It seems like captive-raised individuals could be particularly problematic for the 'population range expansion' dataset, since we don't know if the populations diverge from each other due to genetic differences or phenotypic plasticity.

RESULTS

SC 30; Line 133: "In the separate analyses" – it would reduce the readers' cognitive load if you remind them in very clear language what you mean, given that you have several things going on in this first meta-analysis.

SC 31; Line 136: What do you mean by "relatively strong positive effect"? In Figure 2, the interval for locomotor capacity appears to cross 0, if just barely (so maybe there's an effect, but can you really say it's a relatively strong one? Relative to what?) Also, I notice the interval is a "credible interval" rather than a "confidence interval." Many (maybe most?) of your readers won't be well-versed in Bayesian statistics, so you may need to provide further explanation. What is a credible interval, and do we interpret it the same way we would a confidence interval? Perhaps give that explanation around line 127, when you first mention credible intervals.

SC 32; Line 138: It would reduce the readers' cognitive load if you remind them what types of traits are in the "locomotor capacity" and "metabolism" categories. They may have forgotten that "locomotor capacity" and "metabolism" contain multiple traits lumped together (or they may not have even read the methods). The next sentence touches on some traits, but you could edit the start of the paragraph to be more explicit/clear.

SC 33; Line 141: Ditto the earlier comment about credible intervals – it would help most readers if you provide at least a little explanation, since most are familiar only with confidence intervals. What do naïve readers need to know to be able to interpret the credible intervals?

SC 34; Line 154 (and perhaps throughout subsection): Give the reader some indication that numbers in parentheses and brackets represent estimates and 95% credible intervals.

SC 35; Lines 157-159: You don't report the estimate and credible interval for time since divergence (and I notice the credible interval overlaps 0, even if just barely).

SC 36; Lines 162-163: It seems a bit contradictory to indicate that there's no evidence of publication bias in one sentence, and then to say "After accounting for publication bias..." Is it contradictory, or am I missing something? (And if I am missing something, keep in mind that other readers may also miss something, as it is currently written.)

SC 37; Lines 164-165: Is there a difference between "greatest confidence in effect size" and "greatest effect size"?

DISCUSSION

SC 38; Line 181: "...physiological traits *considered by existing studies* have only limited influences..." Who knows if some trait we haven't measured has a big influence?

SC 39; Line 185: It seems like the detailed discussion of metabolic rate should be its own paragraph.

SC 40; Line 200: It's a little strange to start the paragraph with "In contrast, maximal metabolic rates can constrain movement" when the previous paragraph pointed out that maximal metabolic rate did not predict movement (line 185), even though there's not necessarily a discrepancy between the two statements. Can you improve the logic flow here so that readers don't get confused about the results of your meta-analysis vs. empirical findings of individual papers? Maybe with a different transition phrase or topic sentence?

SC 41; Line 225: Typo – "such as"

SC 42; Lines 243-248: Having not read the papers included in the analysis, I have a lot of questions about things that could cause the 'dispersal to warmer environment' trend that might change the conclusions drawn. For example, I know that for some studies, you used temperature data published by the original authors, and for others, you used temperature data pulled from the Global Climate Extractor (lines 345-350). I could imagine scenarios where the temperature used for one part of the range was published 20, 50, or more years ago, and the temperature used for the other part of the range reflects a more recent recording, so they are not really comparable due to climate change. If it so happens that range core temperatures more often reflect old data and range edge temperatures more often reflect recent data, then you could spuriously get the result that species are expanding into warmer environments. One way around this might be to use coordinates to extract climate data for all sites instead of using the study's measurements for some of them. At the very least, the authors might spend more time chewing on this result and expanding the discussion to explore it (including its nuances and limitations) a little further, especially given broad interest in any result that could have implications for climate change.

SC 43; Lines 249-265: The question of genetic determination vs. phenotypic plasticity nagged at me from the beginning of the paper. I'm glad the authors have addressed it in the discussion (and it strikes me as a very interesting discussion)! I would encourage you to also allude to it earlier, definitely in the introduction (perhaps in the paragraph that starts on line 100) and maybe also in the

abstract, so that readers know from early on that you have indeed thought about it. Beyond just mentioning phenotypic plasticity in an earlier section of the paper, you might consider commenting there on the implication of plasticity for the conclusions that one would draw from the meta-analyses.

SC 44; Line 252: Typo – “diversion” -> “divergence” ?

SC 45; Line 252: The word “exacerbate” implies a judgement that genetic differences are inherently a bad thing.

SC 46: You make no mention in the discussion of differences among taxonomic groups, or the potential consequences of some groups being underrepresented in the existing literature – I wonder what your thoughts are. The omission seems especially strange given that you mention taxonomic differences in the abstract, as if it were a major finding.

FIGURES

SC 47; Figure 3. The headings “metabolism” and “locomotor capacity” cause confusion because they do not align with your convention in Figures 2 and 4, where the heading on top of the plot indicates the thing being predicted (activity, exploration, or dispersal). I can think of two alternate solutions to prevent confusion: 1) Move the “metabolism” and “locomotor capacity” headings to the left so that they are aligned over the different measures of metabolism and locomotor capacity, and then place a new heading along the lines of “Movement” at the top of the plots. That means the two plots will have the same heading at the top, but at least then the reader will know both plots are showing the effects of various traits on the grouped activity/exploration/dispersal variable. 2) Change the headings to “Effects of metabolism [or metabolic variables, or whatever] on movement” and “Effects of locomotor capacity on movement”.

SC 48; Figure 4. It is not immediately clear what this figure represents. In Figure 2, it’s easy to see that the effect sizes with their intervals represent the effect of locomotor capacity on activity, the effect of condition on dispersal, etc. In Figure 4, the effect sizes with their intervals do not represent the effect of Reptilia on activity, the effect of Aves on Exploration, etc. Right? How does the physiology work its way in here? I’m not sure if it would be best to edit the figure, to explain in the figure caption, or both.

SC 49; Figure 6b: Similar confusion to Figure 4.

SUPPLEMENTARY INFORMATION

SC 50: I expected to find a list of references for the studies included in each meta-analysis, and did not. It seems like such a list would be necessary for replication.

Reviewers' comments:

Reviewer #1 (Remarks to the Author):

The authors should be commended on a robust study, with excellent result presentation and a well written document.

the first analysis to assess the link between physiology and movement, is well established, but I presume was used to demonstrate suitability of the methods used.

1. The second analysis assessed if physiology differed between animals within the core or edge of the species their geographical ranges.

This is perhaps the more interesting analysis but after reading the paper multiple times I could not work out how you were able to compare the core and range effects upon physiology and location from the same species across multiple studies, which I presume used different sampling methods. Where these findings comparable?

This concept of differing physiology around core and edge effects has been demonstrated a few times now in single species, where the sampling methods were comparable. for example cane toads in Australia.

RESPONSE: this is a challenge for meta-analyses, which compare data across studies. The biases due to study (e.g., as a result of different techniques, samples sizes, etc.) are therefore assessed with several formal analyses that can shed light on the extent of that heterogeneity. We present the results of these analyses (heterogeneity; Table S4, and outlier analysis, Fig. S4) in the Supplementary material. As is standard practice, we also corrected for potential study effects by including 'study' as a random effect in the analysis. We also included 'effect size' as a random effect in case there are unusually high or low effect sizes, and 'species' in case any particular species had a disproportionately large effect. Please see the explanation in the Methods (1st paragraph under the 'Individual variation' heading).

Reviewer #2 (Remarks to the Author):

BRIEF SUMMARY OF THE MANUSCRIPT

The authors conducted two separate meta-analyses to link physiology with ecology via animal movement, broadly defined.

The first meta-analysis focused on the link between physiological traits (measures of metabolism, locomotor performance, immunity, etc.) with animal movement (activity in a familiar environment, exploration in a novel environment, and dispersal). It resulted in limited support for effects of some physiological traits on movement.

The second meta-analysis examined whether individuals from the core of their range vs. the edge of their range differ in physiology. It provided evidence for divergence of numerous physiological traits, including some that would be expected to have ecological effects. Additionally, the degree of divergence is positively related to the time since divergence. It is not clear the extent to which

this divergence might be the result of genetic differentiation vs. phenotypic plasticity. This second meta-analysis also explored whether temperature or precipitation might differ systematically between range core and edge, finding evidence for warmer temperatures at range edges (with some caveats/limitations).

OVERALL IMPRESSION

The authors have chosen an interesting and important topic with implications for a variety of fields. I'm always excited to see connections between individual-level traits and population-level phenomena!

1. As with many interesting and important topics, this one presents some inherent challenges (not insurmountable).

First, this work will likely interest readers in disparate fields, including physiology, population genetics, behavioral ecology, and community ecology. Researchers from these fields have very different knowledge bases, and they don't necessarily share a vocabulary. They likely have different ideas of what one usually means by "movement"! Thus, the authors must work hard to make sure they communicate clearly with all parts of the potential audience, which involves clearly defining/explaining terms and ideas that might not need defined/explained in a paper targeting a narrower audience. In my specific comments, I aimed to point out places where the authors could expand or improve their explanations to reach their broad audience.

Second, many researchers who would be interested in these results are not intimately familiar with meta-analyses or Bayesian statistics. As the manuscript is currently written, such researchers will likely have trouble understanding how to interpret some results, and could use more guidance from the authors. In my specific comments, I have indicated some places where the authors could provide such guidance.

The authors can certainly rise to these challenges with some revisions.

RESPONSE We responded to the specific comments as outlined below, and also edited the ms for clarity

Finally, please note that I have never conducted a meta-analysis, nor am I intimately familiar with the mechanics of meta-analyses. I also have only a passing familiarity with Bayesian statistics. Therefore, I cannot evaluate the statistical analyses or the validity of claims based on those analyses. Therefore, the editor should ensure that another reviewer who does have experience with meta-analyses and Bayesian statistics has critically reviewed the methods and results sections.

SPECIFIC COMMENTS

To aid authors in preparing their response, specific comments are numbered consecutively (SC 1, SC 2, etc.), and most refer to line numbers.

TITLE

2. SC 1: The title given in the Supplementary Information document (Does physiology predict animal movement: exploration, dispersal, and invasiveness?) differs slightly from the title given in the main document (Does physiology predict animal movement: activity, exploration, and dispersal?) Either way, the punctuation makes it a bit odd to read, in my opinion. If other reviewers have the same impression, you may consider a changes (e.g., “Does physiology predict animal movement? A meta-analysis of physiological effects on X, Y, and Z” or something along those lines). It also seems to me like the title alludes to the first meta-analysis but not the second; authors can decide if that matters to them.

RESPONSE: we edited the title along the lines suggested by the referee, and we ensured that the title is the same in the main ms and SI (Does physiology predict activity, exploration, and dispersal in animals? A meta-analysis)

ABSTRACT AND INTRODUCTION

3. SC 2; Lines 34-46: Although this paragraph isn't labelled “Abstract,” it reads as one and I'm going to interpret it as such. It sounds like your meta-analyses produced interesting findings, but this paragraph currently doesn't convey enough information for the reader to really “get” the context or importance of your findings – some things don't make sense at their current level of detail (I'll give specific examples below). Remember, the abstract is the only part of the paper that many people will read, so you want to make sure it conveys what's interesting and/or important about your findings (including why they are interesting and/or important). It would help if you drew more connections among ideas.

RESPONSE: we re-wrote the opening paragraph according to the reviewer's suggestions as detailed below. The paragraph cannot exceed 150 words, which means that it has to be pretty compact.

4. SC 3; Line 34 (and throughout): The word “movement” can convey many different ideas, and it's not immediately clear to readers which one(s) you are focused on (moreover, readers from different fields will have different initial thoughts on what you might mean). Locomotion? Dispersal? Home range size? Daily movement distance? It would help to be more precise in your language. You don't need to erase all use of the word “movement” (for example, in lines 58-59 it seems appropriate and well-explained), but every time it appears, consider whether replacing it with a more precise word or phrase would improve communication. In the abstract, in particular, I think some replacements would help. The reader has to get pretty far into the paper before getting an explanation of what exactly “movement” means in the context of your analysis, and even then, it's not as clear-cut as I think it could be (additional specific comments provided below).

RESPONSE: we rephrased the first sentence to define "movement" in the specific context of our analysis.

5. SC 4; Line 39: Similar to the previous comment, when you say “individual movement” do you mean “daily movement distance,” or something else?

RESPONSE: we now use the specific terms (activity, exploration, and dispersal), which we used to define 'movement' in the first sentence

6. SC 5; Line 39: I notice a marked difference in the firmness/confidence of your conclusion in the abstract (“We show that locomotor performance and metabolism affected individual movement”) vs. in the results (“There were positive effects of active metabolic rate, sprint speed, and endurance, although the overlap with zero of the credible intervals reduce the confidence of the latter two results”; lines 139-142) and discussion (“there was quite large variation in their effect sizes so that the confidence in their predictive power was limited”; lines 183-184). You should be careful to avoid overstating your results in the abstract, especially since that is the only part some people will read.

RESPONSE: we re-phrased the results summary to better reflect the level of confidence in our analysis.

7. SC 6; Lines 38-41: You list several traits (locomotor performance, metabolism, individual movement, corticosterone, immunity) that differ between the range core and edge. How do you think these traits are connected, if at all? Do you (or others) hypothesize causal relationships? Correlations due to some underlying factor? Coincidence? If you can include a sentence or two to expand on these ideas, the abstract would become more compelling.

RESPONSE: we now added that these traits may be linked to dispersal success. However, as far as we are aware, there are no experimental studies that test explicitly for a cause-and-effect relationship (e.g., by manipulating immunity and then assessing dispersal relative to a control). Hence, the link to dispersal success is speculative, but not entirely unreasonable, and we discuss this further in the Discussion.

8. SC 7; Lines 41-42: Any idea why physiological differences would be more pronounced in birds and amphibians? Again, the abstract will be more compelling if you can go beyond a raw presentation of your findings.

RESPONSE: As far as I know there is no a priori biological reason why differences should be more pronounced in amphibians and birds compared to other taxonomic groups. However, the taxonomic and geographical coverage in the literature is sparse, so that the available data are unlikely to be representative of whole taxonomic groups. It is more likely that the taxa and areas that happened to be sampled showed these patterns. We mention this now in the introductory paragraph.

9. SC 8; Lines 43-44: The trend for dispersal to occur in the direction of warmer environments has some major caveats/limitations. You point out some in the discussion, and I have some additional thoughts, detailed later in this review. Given these caveats/limitations, I think you should be much more careful about how you present this result in the abstract (and maybe even whether you should present it, especially given that it seems slightly tangential to the core focus of your paper).

RESPONSE: we agree and removed these lines.

10. SC 9; Lines 45-46: Can you be more specific? What traits (or types of traits) and geographic regions are currently underrepresented in the literature?

RESPONSE: the last two sentences now read: "Physiological effect were particularly pronounced in birds and amphibians, and taxon-specific differences may reflect biased sampling in the literature, which also focussed primarily on North America, Europe, and Australia. Hence, physiology can influence movement, but measurements were biased towards metabolic and locomotor performance, and data are too sparse currently to draw general conclusions about geographic or taxonomic difference."

11. SC 10; Line 48: Careful with the word "defines." Although foraging is important for habitat use and interactions within ecosystems, other things matter, too (for example, finding mates or avoiding predators).

RESPONSE: we replaced "defines" with "influences"

12. SC 11; Lines 63-64: "Information they receive from their environment" is just one of many reasons individuals may vary in their tendency to initiate movement. You get more into the initiation of movement in the next paragraph (lines 67-71); some minor re-organizing would help here. Maybe remove this content from the current paragraph and work it into the next?

RESPONSE: we moved this sentence to the end of the next paragraph, which indeed improves the flow.

13. SC 12; Line 64: What exactly do you mean by "movement per se"? Movement, broadly defined? The "per se" makes me wonder if you have something special in mind that I may or may not be getting.

RESPONSE: we changed "movement per se" to "the speed and distance moved" - 'per se' tried to delineate the process of movement from the motivation to move, albeit clumsily.

14. SC 13; Lines 63-66: "Individuals vary in their tendency to initiate movement" ... "Consequently, there is substantial variation between individuals in their tendency to move" – this is a circular. I recommend re-working these sentences (some re-working would probably happen anyway, if you take my suggestion of moving this idea into the next paragraph).

RESPONSE: we removed the circularity and changed the sentence to : "Consequently, not all individuals in a population are likely to disperse¹⁰."

15. SC 14; Lines 74-76: Can you be more specific, maybe by providing an example? This sounds interesting, but not everyone in your potential audience will automatically know what you are talking about. Also, since it is a general statement, you might want to cite one or two more studies beyond just the one on guppies.

RESPONSE: we now provide two extra examples (+ references) and also made the initial general statement specific to guppies to read: "...mismatches between the parental environment and the actual environmental conditions experienced can have negative influences on physiological performance and stimulate dispersal in guppies (*Poecilia reticulata*)¹⁴. Similarly, exposure of mothers to predators increased their daughters' tendency to disperse in the fish *Gambusia affinis*¹⁵. In the spider *Cyrtophora citricola*, the early natal environment influenced dispersal behaviour of offspring¹⁶."

16. SC 15; Line 82: "Movement relies on muscle-powered locomotion" – many plant seeds disperse despite their lack of muscles, and planktonic organisms also move without muscle-powered locomotion. Perhaps specify which taxa/circumstances you are talking about when making such broad statements?

RESPONSE: we now specified: "Swimming, flight, and terrestrial movement such as running and walking rely on muscle-powered locomotion, and ..."

17. SC 16; Lines 85-86: Can you provide some explanation of what you mean by "cardiac scope" for readers who are not physiologists?

RESPONSE: we now define cardiac scope in this sentence: "...and the capacity of the heart to pump sufficient blood to sustain exercise (cardiac scope) may be constrained"

18. SC 17; Lines 88-90: "For example..." I can't put my finger on why, but this statement feels incomplete, as if you stopped partway through a thought (even though it is a grammatically complete sentence). Maybe read it out loud to yourself to see if you get the same feeling (and revise, if you do)?

RESPONSE: I think that the sentence just did not provide enough explanation of the point we wanted to make. I have now rewritten it to: "Physiological characteristics typically vary between individuals within populations, and these differences may impact the tendency and extent of movement. For example, there was a three-fold difference in the metabolic cost of transport (i.e., the energy used to move a given mass for a given distance) among individual zebrafish, which influenced the distance individual moved in an artificial stream ³⁰."

19. SC 18; Lines 91-93: Given that "tendency to move" (in any sense of the word "move") relies on behavior as well as physiology, and behavior is more labile than physiology, then we might expect even more intraspecific variation in "tendency to move" than we see for the physiological traits discussed earlier in the paragraph. (You may or may not want to address this point.)

RESPONSE: we agree, but would prefer not to insert a discussion on physiological vs behavioural impacts, particularly because behaviour relies on neurophysiology to a large extent so that the distinction may not be all that clear. However, we edited this sentence to make it specific to physiology and to not exclude behaviour: "If movement relied on physiological capacities, it may be expected that the variation in physiological characteristics introduces differences in the tendency to move among individuals of the same populations."

20. SC 19; Lines 96-98: "Dispersal success" seems like a subtly different concept from "tendency to move" or "dispersal rates" mentioned earlier in the paragraph (an animal might have a high "tendency to move," but it may not succeed in dispersing, or vice versa, for a variety of reasons that may or may not be related). Re-read to see if you are communicating what you mean to communicate, or if you need to edit or expand on any of these ideas.

RESPONSE: yes, 'rates' is better than 'success' and we changed the sentence accordingly.

21. SC 20; Lines 106-108: Perhaps streamline this sentence by cutting a clause: "Potential differentiation in physiological phenotypes between the core and edges may influence community

ecology through a variety of underlying traits, from disease resistance to social behaviour."

RESPONSE: we changed this sentence as suggested

22. SC 21; Line 113: Here, it would be especially helpful to precisely communicate what you mean by "movement," since this is where you define the scope of your study.

RESPONSE: we specified movement: " ...for animal movement, including activity within familiar environments, exploration of novel environments, and dispersal."

MATERIALS AND METHODS

23. SC 22; Lines 289-292: Individuals of what taxa? All animals? All vertebrates? A reader can currently glean the answer from figures and supplemental materials, but it should also be mentioned in the main text. This would be a good place to refer the reader to Figure 1.

RESPONSE: we now added: "...measure of movement and physiological trait(s) in the same individual of any species that uses muscle-powered locomotion for activity, exploration or dispersal (see also Supplementary methods, data exclusion criteria)."

24. SC 23; Lines 293-304: Is criterion (1) your definition for "individual movement"/"movement by individuals"? If so, can you make that explicit? If not, can you change something about this paragraph or the preceding one to clarify? (The answer to this question was clearer the second time I read the manuscript, but it needs to be crystal clear on the first read.)

RESPONSE: we changed the sentence to: "(1) the study reports measurements of movement by individual animals such as 'activity' and 'exploration'."

25. SC 24; Lines 293-304: You make no mention of how many studies were lab-based vs. field-based. I suspect many (most?) experiments were lab-based, which could have implications for interpretation. Behaviors displayed in lab assays may or may not measure what we think they measure, and they may or may not correlate with behavior in the field. At a minimum, it would be nice for the methods to indicate whether both categories of study were included, and for the discussion to touch on some of the implications for our interpretation of the results, assuming lab-based studies were included. Here's one recent paper on the topic:

Mouchet A. and N.J. Dingemanse. 2021. A quantitative genetics approach to validate lab- versus field-based behavior in novel environments. *Behavioral Ecology* 32:903–911.

RESPONSE: We now added to the Methods: "We included both laboratory and field studies; in the event, most studies measuring 'activity' and 'exploration' were laboratory based (85%), while dispersal was measured primarily in the field (90%).", and to the Results: "Most of the data on activity and exploration in our analysis were derived from laboratory studies. The motivation for movements such as exploration of unfamiliar environments may differ between laboratory settings and the field⁵⁷. Movement speed and underpinning physiological (e.g., metabolism) processes may thereby also differ between field and laboratory setting. There are currently not sufficient field studies on activity and exploration for formal comparisons, and this would be an interesting avenue for future research.", and we cite Mouchet and Dingemanse 2021 in the Discussion (57).

26. SC 25; Lines 305-306: Even if you don't have space to list all measures of physiology in the main text, readers should get some idea of traits considered without having to consult the supplementary material. One possibility would be to say something like "measures of physiology included are given in Supplementary material Table S1; they include traits related to metabolism (15 total), locomotor capacity (6 total)..."

RESPONSE: we now provide these details: "Most effect sizes stem from measures of metabolism (k = 106), locomotor performance (k = 66), and body condition (k = 40), and a full list of physiological measures is given in Supplementary material Table S1."

27. SC 26; Line 332: What do you mean by "recent arrival"? Is this somehow different from a dispersal/invasion front?

RESPONSE: we deleted "recent arrival"

28. SC 27; Lines 335-336: How often do researchers have information on the year when the populations at the range core were established? I would expect this information to be readily available only for species that have been introduced to new locations and then expanded their introduced range. I would be surprised to see this information for a population in the core of a species' native range.

RESPONSE: Only four studies did not provide information on the year when the core range population established. These studies were not used to analyse the influence of number of years diverged and the magnitude of the response (Fig. 5c). For populations expanding from their native range, those studies had historical records when the species previously expanded (which we used as core range). We now included the comment: "Four studies did not report the time of establishment of the core range, which we did not include in this analysis."

29. SC 28; Lines 335-336: It is not clear whether all the studies included in this second meta-analysis were of organisms expanding the size of their introduced range (such as cane toads in Australia), or whether some studies may have been of organisms that are expanding out from their native range (such as cowbirds in North America). It would be nice to know the proportion of each type of study included in the analysis. Since Communications Biology relegates Materials and Methods to tiny text at the end of the paper, you may consider clarifying this point elsewhere, as well.

RESPONSE: We have now included the following sentence in the results section: "Of those studies, 36 examined species expanding from their introduced range, and 7 with species expanding from their native range".

30. SC 29; Lines 354-355: I'm concerned about the inclusion of data from captive-raised individuals, given phenotypic plasticity of many traits. It seems like captive-raised individuals could be particularly problematic for the 'population range expansion' dataset, since we don't know if the populations diverge from each other due to genetic differences or phenotypic plasticity.

RESPONSE: We added that: "this moderator is relevant for the 'individual movement' analysis only, and refers to common laboratory animals such as zebrafish"; no population expansion data exist for captive-raised animals.

RESULTS

31. SC 30; Line 133: "In the separate analyses" – it would reduce the readers' cognitive load if you remind them in very clear language what you mean, given that you have several things going on in this first meta-analysis.

RESPONSE: we edited this sentence to: "Next, we analysed effects of physiological traits on activity in familiar environments, exploration of novel environments, and dispersal separately. Only body condition, locomotor performance, and metabolism had sufficient numbers of effect sizes for analysis."

32. SC 31; Line 136: What do you mean by "relatively strong positive effect"? In Figure 2, the interval for locomotor capacity appears to cross 0, if just barely (so maybe there's an effect, but can you really say it's a relatively strong one? Relative to what?) Also, I notice the interval is a "credible interval" rather than a "confidence interval." Many (maybe most?) of your readers won't be well-versed in Bayesian statistics, so you may need to provide further explanation. What is a credible interval, and do we interpret it the same way we would a confidence interval? Perhaps give that explanation around line 127, when you first mention credible intervals.

RESPONSE: we added the explanation: "note that credible intervals are the Bayesian equivalent of confidence intervals, and are interpreted in the same way." in the paragraph above as suggested, and to the Statistical analysis section in the Methods. We rephrased the sentence describing the results to: "... but there is a high level of confidence (marginal overlap of credible intervals with zero) that locomotor capacity had a positive effect on activity".

33. SC 32; Line 138: It would reduce the readers' cognitive load if you remind them what types of traits are in the "locomotor capacity" and "metabolism" categories. They may have forgotten that "locomotor capacity" and "metabolism" contain multiple traits lumped together (or they may not have even read the methods). The next sentence touches on some traits, but you could edit the start of the paragraph to be more explicit/clear.

RESPONSE: Fig. 3 lists all the different traits and we refer to that figure now earlier in the paragraph; it would be quite messy to list them in the text as well.

34. SC 33; Line 141: Ditto the earlier comment about credible intervals – it would help most readers if you provide at least a little explanation, since most are familiar only with confidence intervals. What do naïve readers need to know to be able to interpret the credible intervals?

RESPONSE: please see above: we now added this explanation.

35. SC 34; Line 154 (and perhaps throughout subsection): Give the reader some indication that numbers in parentheses and brackets represent estimates and 95% credible intervals.

RESPONSE: we added the explanation that the numbers represent credible intervals (e.g., (0.02 [95%CI: -0.01–0.005])).

36. SC 35; Lines 157-159: You don't report the estimate and credible interval for time since

divergence (and I notice the credible interval overlaps 0, even if just barely).

RESPONSE: we edited this sentence to read: "There was high confidence in a positive relationship (credible interval:-0.02-0.14) between the time since divergence and the magnitude...".

37. SC 36; Lines 162-163: It seems a bit contradictory to indicate that there's no evidence of publication bias in one sentence, and then to say "After accounting for publication bias..." Is it contradictory, or am I missing something? (And if I am missing something, keep in mind that other readers may also miss something, as it is currently written.)

RESPONSE: this was badly worded, and we deleted "After accounting for publication bias.."

38. SC 37; Lines 164-165: Is there a difference between "greatest confidence in effect size" and "greatest effect size"?

RESPONSE: We changed this sentence to : "Among individual physiological traits, the traits with the greatest confidence in positive effect sizes (i.e., little or no overlap of credible intervals with zero) were hormone....".

DISCUSSION

39. SC 38; Line 181: "...physiological traits *considered by existing studies* have only limited influences..." Who knows if some trait we haven't measured has a big influence?

RESPONSE: Changed as suggested.

40. SC 39; Line 185: It seems like the detailed discussion of metabolic rate should be its own paragraph.

RESPONSE: Changed as suggested.

41. SC 40; Line 200: It's a little strange to start the paragraph with "In contrast, maximal metabolic rates can constrain movement" when the previous paragraph pointed out that maximal metabolic rate did not predict movement (line 185), even though there's not necessarily a discrepancy between the two statements. Can you improve the logic flow here so that readers don't get confused about the results of your meta-analysis vs. empirical findings of individual papers? Maybe with a different transition phrase or topic sentence?

RESPONSE: We re-worded this sentence to: "It is possible that maximal metabolic rates constrain movement because lower maximal rates can limit energy supply to muscles."

42. SC 41; Line 225: Typo – "such as"

RESPONSE: Corrected

43. SC 42; Lines 243-248: Having not read the papers included in the analysis, I have a lot of questions about things that could cause the 'dispersal to warmer environment' trend that might change the conclusions drawn. For example, I know that for some studies, you used temperature data published by the original authors, and for others, you used temperature data pulled from the

Global Climate Extractor (lines 345-350). I could imagine scenarios where the temperature used for one part of the range was published 20, 50, or more years ago, and the temperature used for the other part of the range reflects a more recent recording, so they are not really comparable due to climate change. If it so happens that range core temperatures more often reflect old data and range edge temperatures more often reflect recent data, then you could spuriously get the result that species are expanding into warmer environments. One way around this might be to use coordinates to extract climate data for all sites instead of using the study's measurements for some of them. At the very least, the authors might spend more time chewing on this result and expanding the discussion to explore it (including its nuances and limitations) a little further, especially given broad interest in any result that could have implications for climate change.

RESPONSE: All studies reporting environmental temperatures measured these during the data collection process for both core range and range front. The Climate Extractor obtained current mean data for both range core and edges. We now added the explanations that: " Note that all studies that reported temperature data measured these at the time of conducting field work." and that: "The earliest study in our data set was published in 2002, and climate change may have caused some divergence in temperatures between then and now."

44. SC 43; Lines 249-265: The question of genetic determination vs. phenotypic plasticity nagged at me from the beginning of the paper. I'm glad the authors have addressed it in the discussion (and it strikes me as a very interesting discussion)! I would encourage you to also allude to it earlier, definitely in the introduction (perhaps in the paragraph that starts on line 100) and maybe also in the abstract, so that readers know from early on that you have indeed thought about it. Beyond just mentioning phenotypic plasticity in an earlier section of the paper, you might consider commenting there on the implication of plasticity for the conclusions that one would draw from the meta-analyses.

RESPONSE: We added the following comments (between **) to the Introduction: "As briefly discussed above, individuals with greater tendency to move may have particular physiological characteristics, leading to core-edge differences. * *It is an interesting and as yet unresolved question whether any putative differences between individuals at the core and those at the edge are mediated genetically or epigenetically, or by a mixture of both* ³⁵ * In a species expanding into novel environments, conditions at the dispersal front may also differ substantially from those of the core distribution. As a consequence, the phenotypes that are successful in environments at the range edges may be different to the most successful phenotype at the core of the distributions;* *these differences may arise because individuals with particular genetic make-ups or greater capacity for plasticity have greater fitness at the range edge* ^{36,37}.*"

45. SC 44; Line 252: Typo – "diversion" -> "divergence" ?

RESPONSE: Corrected

46. SC 45; Line 252: The word "exacerbate" implies a judgement that genetic differences are inherently a bad thing.

RESPONSE: We replaced "exacerbate" with "promote".

47. SC 46: You make no mention in the discussion of differences among taxonomic groups, or the potential consequences of some groups being underrepresented in the existing literature – I

wonder what your thoughts are. The omission seems especially strange given that you mention taxonomic differences in the abstract, as if it were a major finding.

RESPONSE: we now included a paragraph on taxonomic underrepresentation at the end of the Discussion.

FIGURES

48. SC 47; Figure 3. The headings “metabolism” and “locomotor capacity” cause confusion because they do not align with your convention in Figures 2 and 4, where the heading on top of the plot indicates the thing being predicted (activity, exploration, or dispersal). I can think of two alternate solutions to prevent confusion: 1) Move the “metabolism” and “locomotor capacity” headings to the left so that they are aligned over the different measures of metabolism and locomotor capacity, and then place a new heading along the lines of “Movement” at the top of the plots. That means the two plots will have the same heading at the top, but at least then the reader will know both plots are showing the effects of various traits on the grouped activity/exploration/dispersal variable. 2) Change the headings to “Effects of metabolism [or metabolic variables, or whatever] on movement” and “Effects of locomotor capacity on movement”.

RESPONSE: We opted for option 2 and have changed the figure headings to “Effects of metabolism on movement” and “Effects of locomotor capacity on movement”.

49. SC 48; Figure 4. It is not immediately clear what this figure represents. In Figure 2, it’s easy to see that the effect sizes with their intervals represent the effect of locomotor capacity on activity, the effect of condition on dispersal, etc. In Figure 4, the effect sizes with their intervals do not represent the effect of Reptilia on activity, the effect of Aves on Exploration, etc. Right? How does the physiology work its way in here? I’m not sure if it would be best to edit the figure, to explain in the figure caption, or both.

RESPONSE: we edited the caption to clarify the Figure. The caption actually contained an error - we stated that activity, exploration, and dispersal were lumped together, rather than physiological traits - which would have caused confusion.

50. SC 49; Figure 6b: Similar confusion to Figure 4.

RESPONSE: We edited the caption to clarify the Figure.

SUPPLEMENTARY INFORMATION

51. SC 50: I expected to find a list of references for the studies included in each meta-analysis, and did not. It seems like such a list would be necessary for replication.

RESPONSE: All studies were referenced (including link to papers) in the raw data file on Github (<https://github.com/nicholaswunz/dispersal-meta-analysis>), and we have now also provided the references in the supplementary information under the heading “Studies included in the analysis”.

REVIEWERS' COMMENTS:

Reviewer #2 (Remarks to the Author):

The authors have done a good job editing the manuscript, and I look forward to seeing it published!